# Rotational Speed Measurement Based on *LC* Wireless Sensors

**DOI:** 10.3390/s21238055

**Published:** 2021-12-02

**Authors:** Yi Zhou, Lei Dong, Chi Zhang, Lifeng Wang, Qingan Huang

**Affiliations:** Key Laboratory of MEMS of the Ministry of Education, Southeast University, Nanjing 210096, China; 230189121@seu.edu.cn (Y.Z.); dl@seu.edu.cn (L.D.); 220191422@seu.edu.cn (C.Z.)

**Keywords:** rotational speed, inductive coupling, *LC* sensor, wireless sensor

## Abstract

This article presents a method for detecting rotational speed by *LC* (inductor-capacitor) wireless sensors. The sensing system consists of two identical *LC* resonant tanks. One is mounted on the rotating part and the other, as a readout circuit, is placed right above the rotating part. When the inductor on the rotating part is coaxially aligned with the readout inductor during rotation, the mutual coupling between them reaches the maximum, resulting in a peak amplitude induced at the readout *LC* tank. The period of the readout signal corresponds to the rotation speed. ADS (Advanced Design System) software was used to simulate and optimize the sensing system. A synchronous detection circuit was designed. The rotational speed of an electric was measured to validate this method experimentally, and the results indicated that the maximum error of the rotation speed from 16 rps to 41 rps was 0.279 rps.

## 1. Introduction

The accurate and continuous detection of rotation speed is very important in industries since it is a key parameter in the condition monitoring and control of rotating parts such as generators, electric motors, and machine tool spindles. The speed information of the rotating part can reflect whether its working state is normal, so it is essential to measure the rotational speed in real time. Hence, various rotation speed sensors have been developed. The basic technology in such sensors is based on optical reflections, magnetic field variation or charge variation. In optical reflection measurement [1,2], the rotation of the rotating parts causes the wavelength of the grating to change periodically. The rotational speed is obtained by measuring the change period of the wavelength. In the measurement of magnetic field variation [3,4,5], a magnet is usually installed on the rotating parts. When they rotate, the magnet causes a variation in the magnetic field that can be detected by a magnetic sensor, and the variation is converted to a square wave signal. In charge variation measurement [6,7,8,9], electrostatic electrodes are utilized to detect the electrostatic charge on the surface of rotating structures due to relative movement with air. In fiber measurement [10], an all-fiber rotation speed measurement structure based on dual-beam speckle interference has been proposed. In triboelectric measurement [11], a self-powered drill pipe sensor that can measure the rotation speed and direction based on triboelectric nanogenerators has been proposed. However, these measurement methods have their strengths and weaknesses during operation in industrial environments, in terms of accuracy, range, and suitability for applications in a hostile environment. Table 1 lists the specifications of the current rotational speed sensors. For magnetic field variation measurement, the response time of the magnetic sensor to the magnetic field variation is long, and therefore its measurement range is limited. For charge variation measurement, it is difficult to operate in humid environments. Fiber measurement requires complex instruments. This paper presents and demonstrates an alternative approach to detect rotational speed using an *LC* wireless sensor. *LC* wireless passive sensors have been widely developed in industrial applications [12,13,14,15,16,17,18,19,20]. They have the advantages of wireless measurement, low cost, and suitability for hostile environments [21]. In contrast to traditional *LC* wireless passive sensors, where only one *LC* tank is used and the sensing capacitance is measured in response to parameters of interest, the system for measuring rotational speed consists of two identical *LC* resonant tanks, where both *L* and *C* are fixed during the measurement. One tank is mounted on the rotating part and the other, as a readout circuit, is placed directly above the rotating part. Here, series matching capacitors in the readout circuit (i.e., the readout *LC* resonant tank), are added so that the non-distortion of the synchronous detection signal can be obtained [22]. Compared with the methods listed in Table 1, the *LC* synchronous detection method proposed in this article has the proper measurement range for industrial environments (e.g., humid environments). In Section 2, the operation principle of the rotational speed measurement system is presented. In Section 3, ADS software is used to simulate the measurement system. The matching capacitance, the coupling distance, and the effect of component deviations are simulated and optimized. In Section 4, different rotational speeds of an electric fan are experimentally measured. To validate this method, an LED (light-emitting diode) optical method of measuring the rotation speed is also performed and compared with the proposed method. Finally, the maximum measurement rotational speed is estimated.

## 2. Principle of Rotational Speed Measurement

Figure 1a is the schematic circuit diagram of a traditional *LC* wireless passive sensor. Its input impedance is written as [12]:(1)$$Z_{i}=R_{O}+R_{S}+j\omega L_{O}+\frac{M{\left(t\right)}^{2}\omega^{2}}{R_{S}+\frac{1}{j\omega C_{s}}+j\omega L_{S}}$$
where *R_O_* and *L_O_* are the resistance and inductance of the readout coil; *L_S_* and *C_S_* are the inductance and capacitance of the *LC* sensing tank; *R_S_* is the parasitic resistance of the coil; and *ω* is the angular frequency, respectively. *M*(*t*) is the mutual inductance of the two inductors. The traditional *LC* sensors usually use the capacitor as a sensitive unit, but the capacitance sensing principle is not suitable for rotational speed measurements. The rotation of the rotating structure will cause the coupling coefficient between the two inductors to change periodically. Therefore, the method in Figure 1b is used here to extract the change period of the rotating structure to measure the rotational speed. The mutual inductance is given by:(2)$$M\left(t\right)=k\left(t\right)\sqrt{L_{O}L_{S}}$$
where k(t) is the coupling coefficient between the two inductors. It is constant in the traditional *LC* wireless passive sensor. For the rotation speed measurement system, the coupling coefficient k(t) is determined by the position between the readout coil and the sensor coil. Since the readout coil is fixed and the sensor coil rotates at the angular velocity to be measured, their relative positions change with time. Therefore, the coupling coefficient k is a time-dependent function, which is written as k(t). Figure 1c is the schematic diagram of the measurement principle, and Figure 1d is the experimental set-up corresponding to Figure 1c.

Using the following substitutions:(3)$$\omega_{S}=2\pi f_{S}=\frac{1}{\sqrt{L_{S}C_{S}}}$$
(4)$$Q=\frac{1}{R_{S}}\sqrt{\frac{L_{S}}{C_{S}}}\;,$$
where *f_s_* and *Q* are the resonant frequency and the quality factor of the *LC* sensing tank, respectively, the real part *Re*(*Z_i_*), the imaginary part *Im*(*Z_i_*), and the phase *ϕ_Z_* of the input impedance *Z_i_* can be expressed as:(5)$$Re\left(Z_{i}\right)=R_{0}+R_{S}+2\pi L_{0}k{\left(t\right)}^{2}Q\frac{f\frac{f}{f_{S}}}{1+Q^{2}{\left(\frac{f}{f_{S}}-\frac{f_{S}}{f}\right)}^{2}}\;,$$
(6)$$Im\left(Z_{i}\right)=2\pi fL_{0}\left[1+k{\left(t\right)}^{2}Q^{2}\frac{1-{\left(\frac{f}{f_{S}}\right)}^{2}}{1+Q^{2}{\left(\frac{f}{f_{S}}-\frac{f_{S}}{f}\right)}^{2}}\right]$$
(7)$$\varphi_{Z}=arctan\frac{Im\left(Z_{i}\right)}{Re\left(Z_{i}\right)}$$

Near the resonant frequency, the real part reaches a maximum and the phase reaches a minimum [12]. Therefore, in the traditional *LC* wireless passive sensor, the sensing capacitance in the *LC* sensing tank can be measured by the resonant frequency through monitoring the real part or the phase of the input impedance based on the frequency sweep [17]. In the rotational speed measurement, the *LC* sensing tank is mounted on the rotating parts, as shown in Figure 1b. Both *L* and *C* are fixed during the measurement, and k(t)=k(t+T), in which *T* is the rotation period. Therefore, it is necessary to scan the frequency during the rotation, presenting a challenge in readout circuits. In this paper, we use the principle of amplitude modulation. As shown in Figure 1b, an AC voltage source with a single-frequency signal *ω_s_* is applied to the input:(8)$$u_{s}=U_{s}cos\omega_{S}t$$

Then, the output voltage across the series resistor *R_O_* will be an amplitude modulation signal. It is written as [18]:(9)$$\dot{u_{O}}=\frac{\dot{u_{s}}}{\sqrt{{\left(R_{O}+R_{S}+\frac{M{\left(t\right)}^{2}\omega_{S}^{2}}{R_{s}}\right)}^{2}+{\left(\omega_{S}L_{O}\right)}^{2}}}∠\varphi_{Z}$$

Unfortunately, this produces an additional phase *ϕ_Z_* dependent on the rotational speed, causing the demodulated signal to fluctuate with the phase. In order to avoid this problem, the scheme in Figure 1c is proposed. A matching capacitor *C_O_* is added in the readout coil so that the readout and sensing *LC* tanks have the identical resonant frequency:$\;\omega_{O}=\frac{1}{\sqrt{L_{O}C_{O}}}=\omega_{S}$. Under the circuit configuration, the output voltage across the series resistor *R_O_* is currently written as:(10)$$\dot{u_{O}}=\frac{\dot{u_{s}}}{R_{O}+R_{S}+\frac{M{\left(t\right)}^{2}\omega_{S}^{2}}{R_{S}}\;}∠0{}^{\circ}$$

Considering that k(t)=k(t+T), in which *T* is the rotation period, it is clear from Equations (2) and (10) that the rotational speed can be easily demodulated from the output voltage across the series resistor *R_O_*. According to Equation (10), the output signal at both ends of the reference resistor *R_O_* is used as the modulation signal. The change frequency of the mutual inductance M(t) is used for the message frequency, and then the input signal of the signal source in Equation (8) is used as the local oscillation signal. The frequency of the local oscillation signal is used for the carrier frequency. The demodulation method used here is product synchronous detection [18].

## 3. Simulation

Synchronous detection requires that a high-frequency carrier has the same phase as the amplitude-modulated signal. To optimize our design, ADS software is used here to simulate the measurement system. The matching capacitance, the coupling distance, and the effect of component deviations are simulated and optimized. Figure 2 is the schematic diagram of the ADS simulation, in which the readout coil (red) has a fixed position, whereas the sensor coil is rotating (blue). The impedance across the readout coil can be simulated by changing the position of the blue coil according to the rotation track. Table 2 lists the parameters used for simulation.

### 3.1. Matching Capacitance

To observe the effect of matching capacitance on the phase of the impedance, the output signal without the matching capacitance, as shown in Figure 1b, is also simulated for comparison. The *LC* sensing tank coil and the readout coil are set to 16 turns, with an inner diameter of 1.2 cm, a wire diameter of 0.2 mm, and a conductor gap of 0.067 mm. The matching capacitance and *LC* sensing tank capacitance are both set to 20 pF. The reference resistance *R_O_* is set to 1 ohm, and the series resistance of the coils, *R_S_*, is set to 0.3 ohm. The axial distance between the coils is set to 1.7 cm. The turning radius is set to 1.5 cm. The external signal source is taken as a sine wave voltage of $\omega_{O}=\omega_{S}=15.44$ MHz with initial zero-phase. The rotation angle of the *LC* sensing tank relative to the readout coil ranges from 0 rad to π rad with step π/4 rad. The simulated impedance phase as a function of rotation angle is shown in Figure 3.

In order to understand the influence of the matching capacitor on the amplitude of the demodulated waveform, the impedance without and with matching capacitor is respectively given by:(11)$$Z_{nc}=Re+jIm$$
(12)$$Z_{c}=Re$$

Here
(13)$$Re=R_{O}+R_{S}+\frac{M{\left(t\right)}^{2}\omega_{S}^{2}}{R_{s}}$$
(14)$$Im=\omega_{S}L_{S}$$

The change of the demodulated signal amplitude ($U_{c}$ with the matching capacitor and $U_{nc}$ without the matching capacitor) with respect to the real part of the impedance is given by:(15)$$\frac{\left|\frac{dU_{c}}{dRe}\right|}{\left|\frac{dU_{nc}}{dRe}\right|}=\left|\frac{{\left(Re^{2}+Im^{2}\right)}^{3/2}\frac{1}{\sqrt{1+\frac{Im^{2}}{Re^{2}}}}}{Re^{3}-ReIm^{2}}\right|$$

It is clear from Equation (15) that the demodulated waveforms with the matching capacitor are more sensitive to real part changes than those without the matching capacitor. Therefore, the introduction of the matching capacitor can make the demodulated waveform more sensitive to change in mutual inductance.

The solution with the matching capacitor in the readout coil has a smaller impedance phase than that without the matching capacitor. According to the principle of product synchronous detection [21], the excitation source voltage is used as the current seismic signal to be multiplied by the output signals, and a low-pass filter is then used to eliminate the two times frequency component. This requires that the phase of the amplitude modulation signal across the reference resistor is zero.

### 3.2. Coupling Distance

The simulated coupling axial distance ranges from 1.5 cm to 1.9 cm with a step of 0.1 cm, and other parameters are listed in Table 1. Figure 4 shows the amplitude of impedance as a function of rotation angle under different coupling distances. It can be seen from Figure 4 that the amplitude decreases as the coupling distance increases. This means that the coupling distance affects the modulation depth of the amplitude-modulated signal. If the coupling distance is too large, the amplitude change of the modulated signal will be insignificant. Therefore, an appropriate coupling distance is needed in order to observe the obvious amplitude modulation effect.

### 3.3. Component Deviations

In practical measurements, there are parameter errors in components. To simulate the effect of component deviations, the excitation source signal is still taken as $\omega_{O}=\omega_{S}$. The component parameter errors appear only in the *LC* sensing tanks. The capacitance relative error and the inductance relative error are defined as $\delta_{C}$ and $\delta_{L}$, respectively. Figure 5 shows the simulated impedance phase as a function of rotation angle under different component errors. The relative position of the coil is set to change from 0 to π every π/4. The capacitance error in the simulation is set to −2% to 2%, and the error interval is 1%. Figure 5a shows that the input impedance will produce extra phase when there is a deviation between the sensor coil and the readout coil matching capacitor. The inductance error in the simulation is set to −2% to 2%, and the error interval is 1%. Figure 5b shows that the input impedance will produce an extra phase when there is a deviation between the sensor coil and the readout coil inductor. Figure 5 shows that both the capacitance error and the inductance error cause the additional phase of the impedance. As discussed above, the additional phase results in the waveform distortion of the synchronous detection. Therefore, in order to achieve the minimum waveform distortion, the capacitance and the inductance in the *LC* sensing tank and the *LC* readout tanks are selected as equally as possible.

## 4. Experiments and Results

The rotational speed measurement system is shown in Figure 6. In Figure 6a, the black cylinder represents the rotating structure. The sensor coil (yellow) was installed on the rotating structure. At the same time, the LED speed measurement method was used here to calibrate the accuracy of the speed measurement. Therefore, the LED reflective material (blue) was also installed on the rotating structure. The readout coil was composed of an inductor, a capacitor, and a reference resistor to form an LCR series loop. An AC voltage source was used to provide a stable sinusoidal signal to the readout coil circuit, and the signal frequency was the series resonance frequency of the LCR circuit. The voltage signal across both ends of the reference resistor was used as the modulation signal, and the signal from the AC voltage source was used as the carrier signal. Then, the modulated signal and carrier signal were respectively connected to the multiplier. Finally, the low-pass filtering method was used to eliminate the high-frequency carrier in the output signal of the multiplier, and the oscilloscope displayed the demodulated waveform containing the speed information. Figure 6b shows a photo of the experimental set-up. In our experiments, both readout and sensing coils were copper planar loop coils. They were wound by a winding machine with inner and outer diameters of 1.2 cm and 2.0 cm, respectively. After the winding, the insulating electronic glue reinforced the two coils, and the constant capacitors were soldered to the coils. Finally, the *LC* sensing tank was mounted on an electric fan controlled by DC voltage. The resonant frequency of the two *LC* tanks was measured to be 15.960 MHz by a vector network analyzer (N5224 PNA, Agilent). The coaxial distance of the two coils was 1.7 cm. In the measurement system, the analog multiplier was an AD835. The −3 dB frequency of the low-pass filter was 10 MHz. The rotational speed was adjusted by changing the DC supply voltage of the electric fan. The supply voltage increased from 4 V to 12 V by a step of 0.5 V. When the test was carried out, the carrier signal and the measured voltage signal across the reference resistor were first multiplied by a multiplier and then filtered by a low-pass filter to remove the high-frequency carrier.

The measured voltage signal across the reference resistor is shown in Figure 7a for the coaxial coupling distance of 1.4 cm at two different rotation speeds, where the waveform diagrams at the speeds of 9.4 rps and 15 rps are represented by the blue line and red line, respectively. Figure 7b is for the rotation speed 9.4 rps under two different coaxial coupling distances, where the waveform diagrams at the distances of 1.4 cm and 2 cm are represented by blue and red, respectively. Figure 7 shows that the factors affecting the amplitude modulation signal not only included the coupling distance but also the jitter and background noise of the rotating structures. So it was necessary to perform synchronous detection on the amplitude modulation signal, and only the changing waveform of the amplitude was required.

Figure 8 shows the demodulated voltage waveforms in the oscilloscope for different rotation speeds. Since the input impedance $Z_{i}$ of the circuit reached a maximum when the readout coil and the sensor coil were aligned, the voltage across the reference resistor $R_{O}$ reached its minimum value. In the signal waveform after demodulation, the minimum value of the voltage corresponded to the lowest point of the demodulated signal waveform. The time interval between the lowest points of the two signals corresponded to the rotation period. Compared with the results in Figure 7, the high-frequency carrier signal and background noise were well-suppressed. It is obvious that the signal period decreased as the rotation speed increased.

To calibrate our measurement results, the output frequency of the proposed system as a function of the rotational speed, measured by the LED method, is shown in Figure 9. It can be seen that the slope is 1.02 ± 0.006 Hz/rps.

Figure 10 shows the rotational speeds measured by our synchronous detection method and by the LED detection method for different DC voltages of the electric fan, respectively. Since the rotating structure is a DC-controlled electric fan, it cannot display its own speed information in real time, so we used the traditional LED measurement method to verify the accuracy of the synchronous detection measurement results. The maximum error was only 0.279 rps compared to the LED detection method. However, the rotational speed measurement based on the *LC* wireless sensor has the advantages of low cost and suitability for hostile environments. The main specifications for the *LC* rotational speed measurement method are given in Table 3.

## 5. Discussion

The simulation and experimental results indicate that the main factors affecting the demodulation waveform were the component deviations between the readout coil and the sensor coil as well as the distance between them. The component error (the inductors, the capacitors, and the additional magnetic permeability introduced by the magnetic parts to the inductors) between the two coils will produce the additional imaginary part of the input impedance, which results in the phase difference between the carrier signal and the modulation signal, so the demodulated waveform will produce random fluctuations. The coil spacing will affect the size of the demodulated waveform fluctuations, which means that if there is a vibration in the coil’s axis in the experiment, the demodulated waveform will also fluctuate randomly. Therefore, the component error and the axial vibration between the coils should be avoided as much as possible in the experiment. In our experiments, the maximum error of the rotation speed from 16 rps to 41 rps was 0.279 rps. Since metal can shield electromagnetic signals [20], the impact of metal in the measurement environment should be avoided.

## 6. Conclusions

In summary, this paper proposed and demonstrated a rotation speed measurement method based on an *LC* wireless passive sensor. By introducing the same capacitor in the readout loop as the sensing loop, the rotation speed could be easily demodulated by the synchronous detection. Experimental results showed that the method had a high accuracy and could measure high rotation speeds. The results also suggest that rotation speed measurement based on *LC* wireless sensors has the advantages of low cost, suitability for hostile environments, and suitability for non-conducting parts.

## Figures and Tables

**Figure 1 sensors-21-08055-f001:**
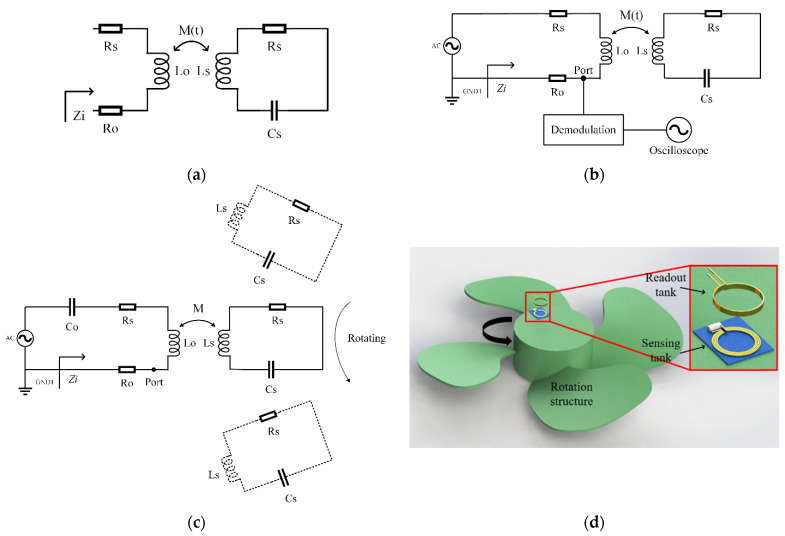
Operation principle of the rotational speed measurement based on a *LC* wireless sensor: (**a**) schematic circuit diagram of a traditional *LC* wireless sensor; (**b**) schematic circuit diagram of a traditional modulation voltage measurement; (**c**) schematic circuit diagram of the readout loop with a matched capacitor; (**d**) 3D modeling of the *LC* rotational speed measuring system.

**Figure 2 sensors-21-08055-f002:**
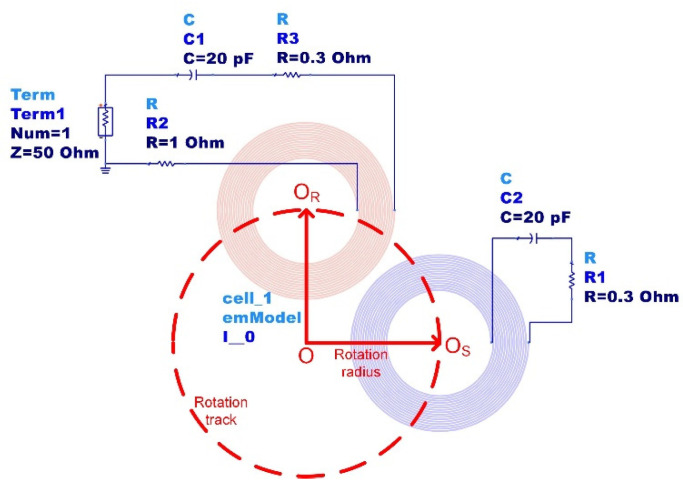
Schematic diagram of the ADS simulation.

**Figure 3 sensors-21-08055-f003:**
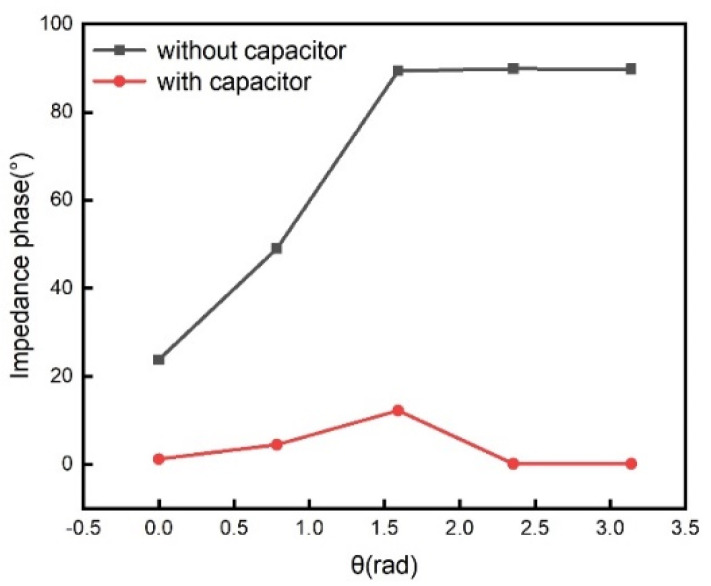
Simulated impedance phase as a function of rotation angle without a matching capacitor (black line) and with a matching capacitor (red line).

**Figure 4 sensors-21-08055-f004:**
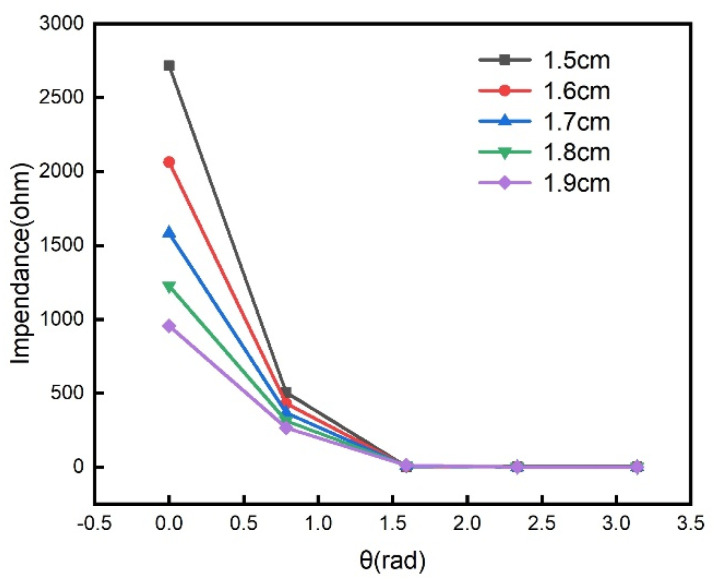
The amplitude of impedance as a function of rotation angle under different coupling distances.

**Figure 5 sensors-21-08055-f005:**
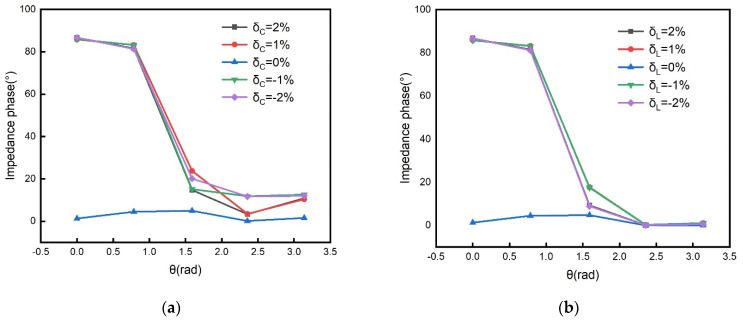
The effect of component deviations on the impedance phase: (**a**) the additional phase of the impedance as a function of rotation angle under different capacitance relative errors; (**b**) the additional phase of the impedance as a func--tion of rotation angle under different inductance relative errors.

**Figure 6 sensors-21-08055-f006:**
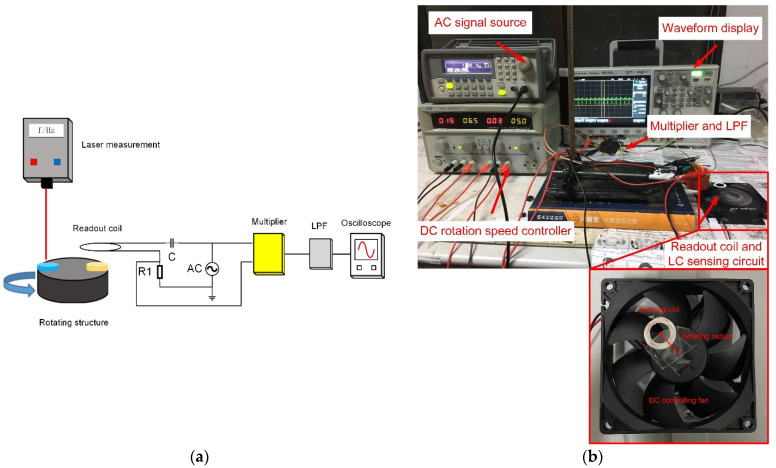
Rotational speed measurement system: (**a**) rotational speed measurement system schematic; (**b**) rotational speed measurement setup.

**Figure 7 sensors-21-08055-f007:**
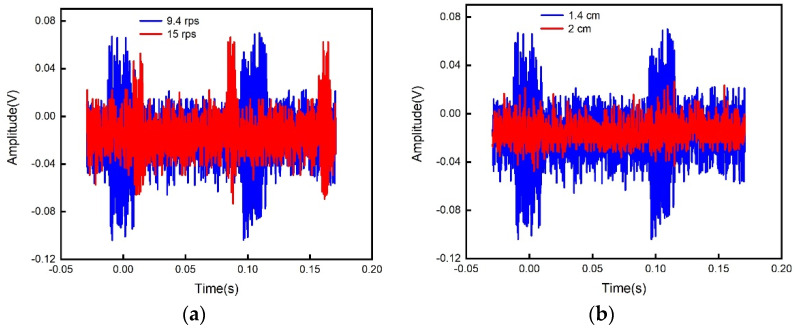
The measured original modulation signal waveform: (**a**) the measured voltage signal as a function of time at different rotational speeds; (**b**) the measured voltage signal as a function of time under different coaxial coupling distances.

**Figure 8 sensors-21-08055-f008:**
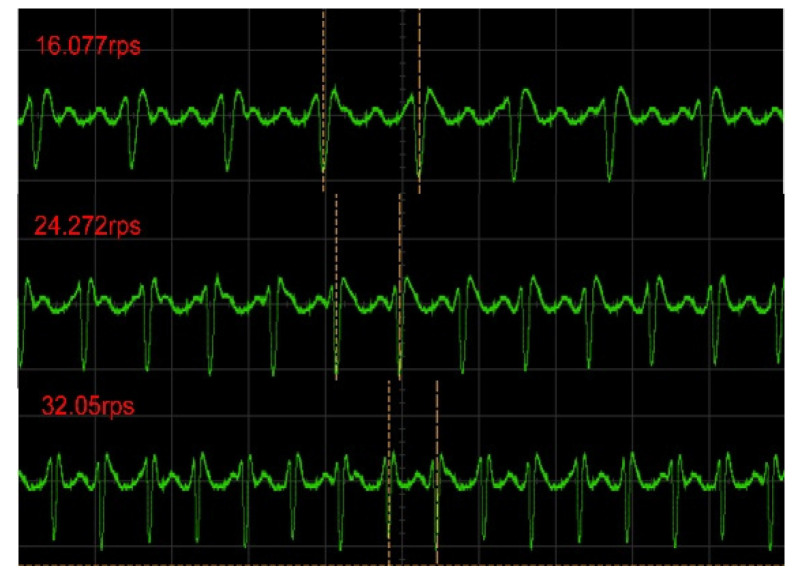
The demodulated voltage signal as a function of time for different rotation speeds in the oscilloscope.

**Figure 9 sensors-21-08055-f009:**
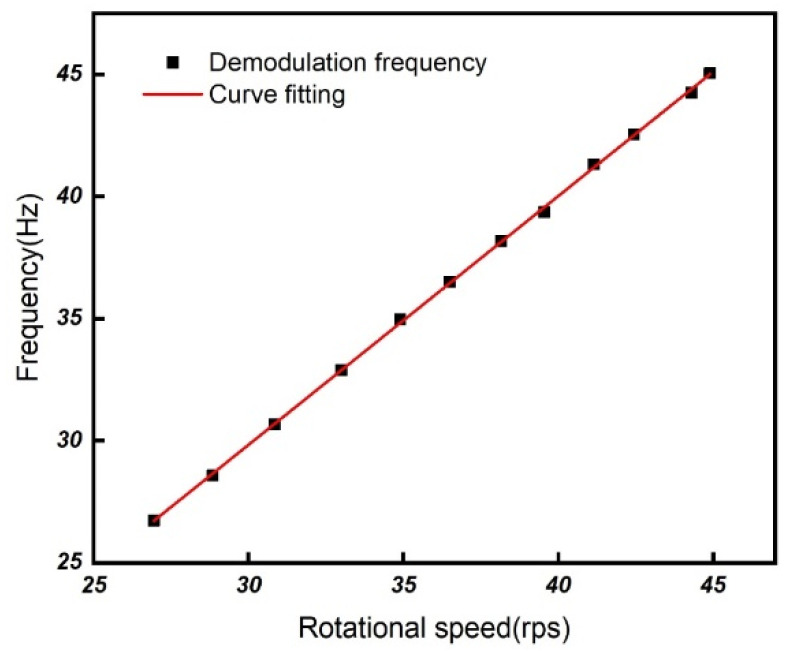
The measured output frequency as a function of rotational speed obtained by the LED detection method.

**Figure 10 sensors-21-08055-f010:**
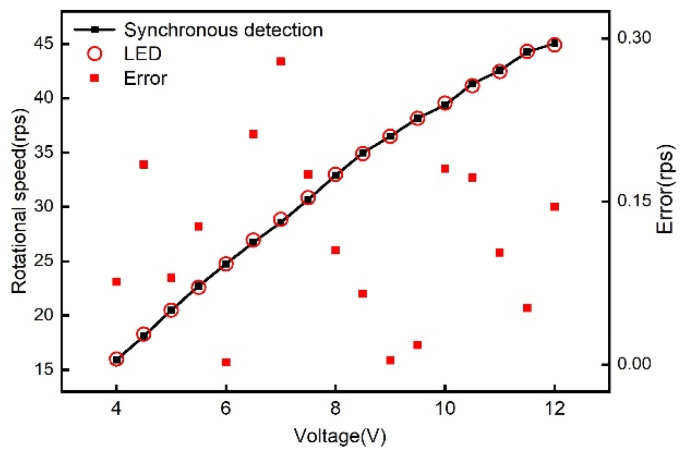
The measured rotational speed of an electric fan under different DC voltages for the synchronous detection and LED detection methods.

**Table 1 sensors-21-08055-t001:** Current rotational speed sensors.

Reference	Year	Measuring Range	Error
Didosyan *et al*. [1]	2003	mrad/s area	/
Wu *et al*. [4]	2016	0 rps~16.67 rps	3%
Wang *et al*. [7]	2015	1.67 rps~50 rps	±1.2%
Li *et al*. [8]	2019	5 rps~53.3 rps	±0.05%
Chen *et al*. [9]	2020	1.67 rps~5 rps	/
Chen *et al*. [10]	2021	7.75 rps~45.68 rps	0.5%
Zhou *et al*. [11]	2021	0 rps~16.67 rps	4%

**Table 2 sensors-21-08055-t002:** Circuit simulation parameters.

Symbol	Quantity	Parameter Value
*L_S_*	sensor inductance	5 μH
*L_O_*	readout coil inductance	4.97 μH
*C_S_*	sensor capacitance	20 pF
*C_O_*	readout coil capacitance	20 pF
*d*	coupling distance	1.7 cm
*R_O_*	reference resistance	1 ohm
*R_S_*	inductance DC resistance	0.3 ohm
*f* _0_	signal source frequency	15.960 MHz

**Table 3 sensors-21-08055-t003:** The specifications of the *LC* rotational speed measurement method.

Measuring Range	Maximum Error	Linearity
16 rps~41 rps	0.279 rps	1.02 ± 0.006 Hz/rps

## Data Availability

Not applicable.

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
