# Peer review of "Rotational Speed Measurement Based on LC Wireless Sensors"

_sensors, 2021, doi:10.3390/s21238055_

Round 1

Reviewer 1 Report

The paper proposes a rotation speed measurement method based on an LC wireless passive sensor, by using two LC circuits. The rotational sensors are intended to be used in rotating parts such as generators, motors, axels and others.

The article provides the principles of rotational speed measurements using the proposed two sets of LC circuits, the equations, a discussion of the matching capacitance and the coupling distance. The article also presents the experimental setup and results.

Page 8 presents the results and a comparison with the LED method. As a suggestion, the authors could provide a comparison table with other methods (similar to Table 1, but including the obtained results of the proposed circuit).

According to the authors, the results suggest that the rotational speed measurement based on the LC wireless sensor has advantages of low cost and suitability for hostile environments.

Minor English review is recommended.

Author Response

Please find our response in the detached file. thanks!

Reviewer 2 Report

The paper presents an idea for sensor of rotation speed based on changes of resonance circuit parameters. The principle is simple and easily implementable at low costs. However in my opinion it requires a deeper analysis and testing. Here are my recommendations:

  1. What is the minimum limit of rotation speed measurement?
  2. Can you derive (11) and (12) in the theoretical section of your paper?
  3. You claim in 205-206, that the maximum rotation speed is estimated to be 38000 rpm but you tested the system only in the order of tens rpm. Can you analyze in simulation or experimentally the behavior of the proposed principal also in wider rpm range?

Author Response

(The authors gave the same response as above.)

Reviewer 3 Report

In my opinion as a reader,

  • the motivation for the work needs to be improvised (please write clearer). 
  • Literature survey needs to be strengthened. It shall not be limited to rotational speed sensors but extend to the wireless LC sensors. There are many recent works on this topic.
  • A photograph of the sensing coil fitted on the rotating part (whole speed is being measured) is needed for better understanding of the implementation.
  • The effect of other objects in the vicinity of the sensing and reader coils needs to be discussed. In most of the rotary systems, like machines, there may be other rotary/moving of fixed magnetic parts.
  • Please specify the material of the rotary part (e.g. blades of a fan). The performance of the proposed system if the blade is conductive needs to be discussed. Limitations, if any should be mentioned in the abstract and conclusion.
  • Fig. 6(a) - is the symbol for the oscilloscope correct? It looks like a source. 
  • Please list the source of errors and limitations of this method. 
  • Performance of this method in comparison with the existing method is recommended. 

Author Response

(The authors gave the same response as above.)

Reviewer 4 Report

The authors have presented a method for detecting rotational speed by LC (inductor-capacitor) wireless sensors using two resonant tanks. The authors should address the following comments for accepting in this high impact journal. 

  1. Table 1 presents the current rotational sensors; however, all of them very old and currently more advance sensors are used in the market.
  2.  The analytical modeling is general and very basic I did not see any novelty and problem related discussion, that how the sensors increase the performance of rotation. Please revise.
  3. The quality of Figure 1 is very poor and description is insufficient, please clearly explain what is the difference among figures a, b and c.
  4. Figure 2 qualities is poor and very simple, which presents no novelty that what the authors have added for enhancing the quality of rotation as compared to current system. It is requested to explain each and every step of Figure 2 and clearly mention the outcomes of proposed framework as compared to current approaches. Please compare your model with current 2021 published rotation mechanisms.
  5.  Figure mentions the results among with and without use capacitor, which is known to everyone that with capacitor the performance will be good and without the performance will be bad. Please clearly describe the simulation results that with the proposed model and without proposed technique how much efficiency is improved.
  6. Figures 4, 5 and 6 seem good but difficult to read, please increase the quality. Figures 5a and 5b are same, and they are not clearly defined.
  7. Again Figure 6 to 7 quality is very poor,  it is requested to elaborate each figure in detail. The text among 190 to 206 make no sense please revise.
  8. Add comparison table which presents the achievements of proposed work.
  9. Add quantitative analysis in the conclusion section and future direction.

Round 2

Reviewer 4 Report

Thanks for addressing the comments in the paper. 

This manuscript is a resubmission of an earlier submission. The following is a list of the peer review reports and author responses from that submission.